# Synergistic Effect of Cerium Oxide for Improving the Fire-Retardant, Mechanical and Ultraviolet-Blocking Properties of EVA/Magnesium Hydroxide Composites

**DOI:** 10.3390/ma15175867

**Published:** 2022-08-25

**Authors:** Jose Hobson, Guang-Zhong Yin, Xiaoli Yu, Xiaodong Zhou, Silvia Gonzalez Prolongo, Xiang Ao, De-Yi Wang

**Affiliations:** 1IMDEA Materials Institute, C/Eric Kandel, 2, 28906 Getafe, Spain; 2Escuela Politécnica Superior, Universidad Francisco de Vitoria, Ctra. Pozuelo-Majadahonda Km 1,800, 28223 Pozuelo de Alarcón, Spain; 3State Key Laboratory of Baiyunobo Rare Earth Resource Researches and Comprehensive Utilization, Baotou Research Institute of Rare Earths, Baotou 014030, China; 4Materials Science and Engineering Area, Escuela Superior de Ciencias Experimentales y Tecnología, Universidad Rey Juan Carlos, Calle Tulipán s/n, 28933 Móstoles, Spain

**Keywords:** ethylene-vinyl acetate, flame retardant, rare earth, mechanical properties, cerium oxide

## Abstract

Rare earth oxide particles have received important attention in recent years, and due to the wide diversity of promising applications, the need for this kind of material is predicted to expand as the requirements to use the current resources become more demanding. In this work, cerium oxide (CeO_2_) was introduced into ethylene-vinyl acetate (EVA)/magnesium hydroxide (MDH) composites for enhancing the flame retardancy, mechanical properties and anti-ultraviolet aging performance. The target EVA/MDH/CeO_2_ composites were prepared by extrusion and injection molding, and the effects of the addition of the CeO_2_ were explored by thermogravimetric analysis (TGA), Differential Scanning Calorimetry (DSC), X-ray Diffraction (XRD), limiting oxygen index (LOI), UL-94, cone calorimetry test, and anti-ultraviolet aging test. Typically, the incorporation of the CeO_2_ allows a significant increase of the elongation at break and Young’s modulus compared with EVA/MDH by 52.25% and 6.85%, respectively. The pHRR remarkably decreased from 490.6 kW/m^2^ for EVA/MDH to 354.4 kW/m^2^ for EVA/MDH/CeO_2_ composite. It was found that the CeO_2_ presents excellent synergism with MDH in the composites for the anti-UV properties in terms of mechanical properties preservation. Notably, the combination of CeO_2_ with MDH is a novel and simple method to improve the filler–polymer interaction and dispersion, which resulted in the improvement of the mechanical properties, flame retardancy and the anti-ultraviolet aging performance of the composites.

## 1. Introduction

Ethylene-vinyl acetate copolymers are widely used in the industrial field, due to their physical and chemical properties. The EVA thermoplastics containing different vinyl acetate contents can be selected in different areas such as transport, electrical, building and electronics, where the flexibility and processing characteristics of these materials are extensively applied to produce hot-melt adhesive, flexible pipes, battery adhesive film or toys, and especially in the cable industry as an excellent insulating material with good physical and mechanical properties [1,2,3]. The typical disadvantage of these polymers is their high flammability; owing to their chemical structure and composition, they can burn easily, and this issue is a major restriction for their application. The most effective way to overcome this drawback is to add flame retardants to the EVA, and recently much research work is focused on the use of environmentally friendly halogen-free flame retardant additives [4,5]. The flame-retardant polymeric materials with low emissions of heat and smoke-suppression effects have become a potential trend. To achieve the required fire resistance grade, aluminum hydroxide or magnesium hydroxide are added into the EVA polymer matrix. These materials have the ability to act in the gas phase, following an endothermic decomposition reaction and causing the dilution of the combustible gases by releasing water during the combustion stage. They also generate a metal oxide coating in the condensed phase, which acts as an insulating layer during combustion [6,7]. Because of its simple, cost-effective, and straightforward fabrication method, as well as its exhibited aptitude for surface functionalization, magnesium hydroxide is selected as candidate in many different research explorations [8]. However, the disadvantage of low efficiency, requiring more than 50 wt.% to reach the flame-retardancy requirements, often leads to the deterioration of physical and mechanical properties of the polymer composites.

In addition to flame-retardant safety, anti-ultraviolet and other thermal-oxidative aging properties are also crucial to improving the comprehensive performance and life of materials. According to reports, CeO_2_ is one of the most attractive rare earth metal oxides. It has several applications in the field of corrosion prevention, electrochemical cells, electromagnetic shielding, thermal coatings, optical and photoelectrical properties [9]. Also, CeO_2_ has great attention because of its unique features like nontoxicity, biocompatibility, oxygen storage capability, optical, and thermal properties, which have significant applications in solar cells, gas sensors, biosensors [10,11], UV shielding [12,13] and flame-retardant additives in polymers [14,15,16]. For typical examples, CeO_2_ was investigated as an additive for in-situ preparation of TiB_2_/Al composite using an exothermic reaction process via K_2_TiF_6_ and KBF_4_ salts [17]. It is reported that by introducing a rare earth oxide material CeO_2_ into the lamella structure of g-C_3_N_4_, the carbonizing level and the chemical flame-retardant inhibition of g-C_3_N_4_ could be enhanced [18]. The growth of rare earth elements into technology advancement, ecology, and economic domains has resulted in a major increase in global demands. Over the next decade, global demand for cars, electronic goods, energy-efficient lighting, and catalysts is predicted to surge. In order to encourage future advances in this field it is necessary to perform further studies for its applications, and flame-retardant synergistic effect is one of them [19]. Cerium oxide serves as an excellent catalyst for organic reaction, oxidant, and so on, and it is expected to have excellent synergistic effects. Furthermore, the mechanism of adding catalysis is complicated and needs further research [9]. CeO_2_ has been also widely used as the reinforcement filler in polishing solution, glass and catalytic converters [20].

Improving the mechanical properties of metal matrix composites by the addition of a ceramic phase is one of the present’s studying hotspots. Understanding the morphology evolution rule of the ceramic phase during laser cladding is a critical issue to its wider industrial applications. The compatibility effects of flame-retardant fillers for producing uniform dispersed polymer composites have been previously studied, and by using phosphorous-based polypyrrole nanoparticles it was possible to obtain significant improvements in the thermal stability and flame resistance of acrylonitrile-butadiene-styrene composites. The synergistic flame-retardancy effect of polypyrrole nanoparticles and their dopant on charring ability was investigated. The rate of burning of the new polymer nanocomposites was significantly reduced to 7.6 mm/min compared to 42.5 mm/min for virgin polymer. Also, peak heat release rate (pHRR), total heat release (THR) and other combustion properties were greatly reduced. This is in conjunction with the suppression of toxic gases emission [21]. Recently, some spherical inorganic nanoparticles and nanotubes have been used in textile treatment. This is to achieve comfortable properties in textile materials. Graphite platelets, ammonium phosphate and N-[3-(trimethoxysilyl) propyl]-ethylene diamine were uniformly dispersed in a commercial binder. The mass ratio of ammonium phosphate was optimized. TiO_2_ nanoparticles of an average size of 20 nm were dispersed in the coating layer and their effect was investigated. Thermal stability for developed textile fabrics was enhanced. The flame retardancy of coated samples was significantly improved with maintaining good mechanical properties [22].

Considering the low overall efficiency of MDH, the catalytic carbon-forming properties of CeO_2_ itself and its excellent UV-shielding properties, together with a lack of systematic studies on MDH-reinforced EVA with rare earth CeO_2_ addition, in this study, varied amounts of the rare earth oxide CeO_2_ were used to modify the EVA/MDH composites. The characteristic changes in microstructure, flame-retardant performance, mechanical properties and UV-aging properties were explored as a function of the CeO_2_ content.

## 2. Experimental

### 2.1. Materials

Ethylene-vinyl acetate (EVA) polymer, (containing 28 wt.% vinyl acetate) with a density of 0.951 g/cm^3^ (25 °C), melt flow index of 4 g/10min (190 °C/2.16 kg) was purchased from DuPont Company (Wilmington, DE, USA), with the commercial name of Elvax 265. Magnesium hydroxide (99.8% purity) with average diameter of about 1 μm was supplied by Liaoning Jinghua New Material company (China). Cerium oxide (CeO_2_, 0.82 μm) was supplied by Baotou research institute of rare earths.

### 2.2. Mixing of Magnesium Hydroxide with CeO_2_

To obtain the modified magnesium hydroxide powder, first, 1000 g of magnesium hydroxide were dried at 110 °C for 6 h to eliminate absorbed water, then the MDH and the cerium oxide powders were added into a high-speed mixer (SHR-10A, Zhangjiagang, China). The temperature and stirring rate of the mixer were set to 100 °C and 2800 rpm. When the set-up temperature was reached, the mixing was kept continuously for 10 min.

### 2.3. Processing of the EVA Composites

All the EVA composites were processed under the same conditions, by hot-melt extrusion of the modified filler into the polymer matrix using a twin screw extruder (Brabender KETSE 20/40) to make the composite polymer granules at 180 °C. In this case the polymer extrusion was made using a 55 wt.% of fillers. The samples for the fire tests and mechanical properties test were obtained by using an injection machine (Arburg Allrounder). The formulations of the samples are given in Table 1.

### 2.4. Characterizations

Thermogravimetric analysis (TGA) was performed with a TA thermogravimetric analyzer (Q50, New Castle, PA, USA) from 0 to 800 °C, with a heating rate of 10 °C/min in nitrogen atmosphere. Particle size analysis was performed in a Bettersizer ST, Laser Particle size analyzer (BT-9300 ST). X-ray diffraction (XRD Malvern Panalytical B.V, Almelo, EA, The Netherlands) was carried out on a Panalytical Empyrean diffractometer (Malvern Panalytical B.V, Almelo, EA, The Netherlands) with Cu Kα radiation resource (λ = 0.154 nm) and Cu filter. Limiting oxygen index (LOI) was obtained using an oxygen index meter (FTT, East Grinstead, UK) according to American Society for Testing and Materials (ASTM) D2863-77 standard. The size of the samples was 130 × 6.5 × 3 mm^3^. The vertical burning test was determined with the UL-94 vertical flame chamber (FTT, East Grinstead, UK) according to ASTM D3801 standard. The size of the samples was 130 × 13 × 3 mm^3^. The fire behavior of the samples was determined on a cone calorimeter (FTT, East Grinstead, UK) according to the ISO5660 standard, under a heat flux of 50 kW/m^2^, using a sample size of 100 × 100 × 3 mm^3^. The scanning electron microscope (Helios NanoLab 600i, FEI, Portland, OR, USA) was used for the cross-section and char residue inspections at a voltage of 5.0 kV and 0.69 nA of current. The samples were coated with a conductive layer of gold before SEM observation, and the fractured surface samples were immersed in liquid nitrogen before the gold sputtering.

Tensile testing was performed on a universal electromechanical testing machine (INSTRON 3384, Norwood, MA, USA) according to ASTM D 638 standard at a test speed of 50 mm/min, and with a load cell of 2000 N. UV radiation was performed in a Dycometal cck-125 climatic chamber irradiance at 40 °C and 35% RH for 0, 24, 48, and 72 h, respectively. The optical microscope Olympus BX51 was used for the surface morphology inspection of the samples after the UV exposure.

## 3. Results and Discussion

### 3.1. Samples Preparation and Filler Dispersion

#### 3.1.1. TGA and Particle Size of the Fillers

The TGA test was performed for the pure MDH, CeO_2_ and for the modified MDH fillers with 3 wt.% and 5 wt.% of CeO_2_; the obtained results are presented in Table 2 and the corresponding curves were plotted in Figure 1a,b. There was no decomposition observed for the CeO_2_ until 800 °C in the TGA test, while the pure MDH showed an initial decomposition temperature of 312.4 °C with a yield of decomposition products of 68.4% at 700 °C, corresponding to the formation of stable MgO [23,24]. After the addition of the CeO_2_, the initial decomposition temperature of the flame-retardant system was increased until 317.5 °C and 324.3 °C for the fillers with a CeO_2_ content of 3 wt.% and 5 wt.%, respectively, and the charring capacity was also improved by promoting the formation of a more continuous and more compact barrier layer, obtaining an increase until 71.9% in the residue. Therefore, a suitable amount of CeO_2_ can provide higher thermal stability and char-forming behavior to the MDH. This decomposition behavior is in agreement with the TGA results discussed elsewhere [9].

The particle size analysis was performed to obtain the standard dimensions of the flame-retardant fillers; the results are revealed in Table 3; the corresponding curves were plotted in Figure 1c. The additive dispersion through the EVA polymer matrix is a fundamental property for this research work, due to the existence of a high proportion of inorganic particles. The results indicate that the pure MDH has a standard size of 0.98 *μ*m, while the CeO_2_ was sized at 0.82 *μ*m; the influence of the rare earth on the MDH flame retardant indicates that for the fillers containing 3 wt.% of the CeO_2_ the particle size obtained was of 0.96 *μ*m, and for the fillers containing 5 wt.% of CeO_2_ there was a 0.94 *μ*m particle size obtained (Figure 1d).

#### 3.1.2. Morphology Analysis by SEM

In order to compare the influence of CeO_2_ on the comprehensive properties of EVA/MDH composites, we designed four samples as shown in Table 1. The composites are obtained by twin-screw extrusion, and then all test specimens required to be tested are obtained by injection molding. The preparation process is shown in Figure 2a. The SEM cross-section was performed to study the additive dispersion within the polymer matrix, and the evaluation indicates that the pure EVA sample has a smooth and continuous fracture surface (Figure 2b). After the 55 wt.% of pure MDH was added into the EVA, the fracture surface of the composite shows the good dispersion of the filler through the EVA matrix (Figure 2c). While due to the existence of a high proportion of inorganic particles, samples EM, EMC3 and EMC5 all show irregular and rough cross-section morphology. In particular, we found a significant crack in Figure 2c, which will lead to the low mechanical properties of the sample to a certain extent. The further addition of the 3 wt.% and 5 wt.% CeO_2_ with a particle size of 0.82 µm can help to form a more uniform surface (Figure 2d,e) with smaller gaps between the fillers and the polymer matrix, and finally improve the performance of the mechanical and flame-retardant properties of the composites, which will be provided in Section 3.4.

#### 3.1.3. X-ray Diffraction (XRD)

XRD analysis was performed to reveal the diffraction peaks of the pure and modified EVA composites; the XRD curves are plotted in Figure 3. The pure EVA sample has two diffraction peaks at 21.2° and 23.5°, corresponding to the (110) and (002) atomic crystal planes of EVA. The sample containing EVA and the unmodified MDH also has peaks at 18.7°, 38.1°, 50.9°, 58.8°, 62.2° and 68.3°, which fit exactly into the diffraction of the MDH crystals at (001), (101), (102), (110), (111), and (103), in accordance with the literature. [25] In this case, the samples containing cerium oxide exhibited additional peaks at 28.80°, 33.10°, 48.10° and 56.80° that correspond to the (111), (200), (220) and (311) crystal planes of the rare earth oxide [26,27].

### 3.2. The Thermal Performance of the Composites

#### 3.2.1. DSC Analysis

The DSC results are shown in Table 4 and the corresponding curves are presented in Figure 4a. All the samples presented the traditional endothermic and exothermic peaks and the thermal properties were recorded from the second heating run. The glass transition of −28.2 °C was observed for pure EVA with a melting temperature of 75.1 °C and a melting enthalpy of 23.2 J/g related to the slightly crystalline phase of the EVA, as previously reported in the literature [28]. No significant changes were observed in the glass transition and melting temperatures after the addition of the pure MDH and modified MDH with CeO_2_ filler, however, a decrease on the melting enthalpy was obtained for the EVA composites; the pure MDH composites showed a melting enthalpy of 13.9 J/g and the samples containing cerium oxide have a lower enthalpy of 11.2 and 11.1 J/g, indicating that the addition of the rare earth is leading to the absorption of lower amounts of heat during the melting cycle of the composites. We can understand that the possible crystallization of EVA is inhibited for pure MDH modified composites due to the introduction of impurities in MDH. Therefore, the enthalpy decreased significantly. With the introduction of CeO_2_, it shows that the good dispersion of CeO_2_ can inhibit EVA crystallization to a greater extent.

#### 3.2.2. TGA Analysis of the EVA Composites

The thermal stability of the EVA composites was measured by the TGA in a nitrogen atmosphere, the summarized data is presented in Table 5, and the related curves are plotted in Figure 4b. The EVA composites exhibited a two-step decomposition process. The first step involves the dehydration of MDH and the loss of acetic acid in EVA at the range of 300–400 °C [23], whereas the second degradation step at the temperature range of 400–550 °C is due to the degradation of ethylene-based chains and to the volatilization of residual polymer [29]. Compared with the pure MDH samples, there is a small shift observed in the T-5% for the samples with cerium oxide; it can be seen that the T-5% of the EM was 327.4 °C, while the EMC3 and EMC5 samples have a lower T-5% of 324.4 °C and 319.1 °C, respectively. In contrast, the char residue at 700 °C was significantly improved; the EM sample has a final weight of 32.5%, while the EMC3 sample with 39.8% and the EMC5 sample had a final char weight of 39.2%.

In addition, as shown in Figure 4c,d, the decomposition in the first stage is advanced, and the maximum thermal decomposition rate in the second stage decreased (Figure 4d). All these show that the introduction of CeO_2_ can significantly change the thermal decomposition path of the composite, so it has more significant char-forming ability. Specifically, as listed in Figure 4b and Table 5, when CeO_2_ is present, it has significantly higher char residue than that of the EM sample.

### 3.3. Fire Safety Performance

The effect of MDH, MDH/cerium oxide on the flammability of EVA was studied by limiting oxygen index (LOI) and UL-94 vertical burning test (Table 6). The LOI value of pure EVA was only 18.2% and it failed to pass UL-94 rating. When 55 wt.% MDH was added, the LOI value increased to 40.9% and the composites passed the UL-94 V-0 rating. After 3 wt.% of the CeO_2_ was added to the EVA system, the LOI value slightly decreased to 38.5% while keeping the UL-94 V-0 rating. After the addition of 5 wt.% of the CeO_2_ the LOI result was 39.9% and the UL-94 rating was V-0. The addition of the CeO_2_ caused a slight decrease in the LOI value; however, the CeO_2_ functionalized flame-retardant composites both presented high efficiency and at this loading of fillers the composites were still able to reach the V-0 rating according to the UL-94 standard.

To investigate the flame-retardant effect of the modified MDH on the EVA composites, cone calorimeter tests were performed at an external heat flux of 50 kW/m^2^. The combustion data, including heat release rate (HRR), total heat release (THR), total smoke production (TSP), and residue, are summarized in Table 7, and the curves of HRR, TSP and char residue versus time for the EVA composites are presented in Figure 5.

From Figure 5a, it can be seen that the heat release rate (HRR) curve of pure EVA has a sharp peak, which indicates the furious burning and heat release. This result is caused by the free combustion of hydrocarbon chains. [4] When the 55 wt.% of MDH was introduced, peak heat release rate (pHRR) values of the resulting composites were considerably reduced. The decomposition of the modified MDH can limit the access of oxygen to the internal layer of material, causing a heat dissipation effect known as “heat sink” during the first stage of decomposition; MDH can also release water molecules while the metal oxide barrier is formed [30].

The most interesting effect of the incorporation of the CeO_2_ is the decrease of the heat release. In fact, the incorporation of even a relatively low amount (3 wt.%) results in the reduction of the pHRR of about 14% (from 491 kW/m^2^ for EM to 425 kW/m^2^ for EMC3). Moreover, with the addition of 5 wt.% of the CeO_2_, the pHRR reduction is higher (28%).

The interesting synergy between the rare earth filler and the MDH has also produced the smoke suppression effect in the EVA system; the composites containing the modified fillers have reduced the smoke production of the EVA composites from 8.6 m^2^ until 7.9 m^2^ (Figure 5b). As shown in Figure 5c, we can find that with the addition of CeO_2_, the char residue increased, which is in good agreement with the results of TGA tests. Furthermore, the modified EVA composites present significant increased TTI as shown in Figure 5d. Both the HRR and THR, of all of flame-retardant samples containing MDH/CeO_2_, showed an evident decrease during burning (Figure 5e,f). All the above indicate that the MDH/CeO_2_ fillers can improve the fire safety of the composites, and the samples with incorporation of CeO_2_ performed much better than those with MDH only.

The fundamental role of CeO_2_ in the flame-retardancy enhancements is assigned to the yielding of a larger amount of solid inorganic residues that serve as a barrier to heat and mass transfer between the pyrolysis zone and the underlying polymer [31]. Typically, as shown in Figure 6, the char residue of the composite increased significantly. In addition, we can clearly see that the introduction of CeO_2_ makes the residual char denser and continuous. Neither EMC3 nor EMC5 have significant micro-cracks compared to sample EM (Figure 6c). Thus, a suitable amount of CeO_2_ plays a critical role in the flame-retardancy and char-forming behavior of EVA/MDH and promotes the formation of more compact char structures in the EVA/MDH composites, which is in good agreement with the literature elsewhere [9].

The effective heat of combustion (EHC), defined as the ratio of heat release to mass loss at a certain time during combustion, can exactly reveal the burning degree of volatiles in gas phase [32,33,34,35]. As shown in Table 7, average EHC (AEHC) was decreased from 35.9 MJ/kg for EVA to 28.6 MJ/kg for EM, and further decreased to 26.3 and 21.8 MJ/kg with the addition of 3 and 5 wt.% of cerium oxide, respectively; the enhanced EHC performance was attributed to the increase of concentration of noncombustible compounds in the gas phase and the higher char residue formed by the inorganic filler’s action. As a result, the heat and mass transfer between the combustion phases was delayed by the imparted flame retardancy. By using these enhanced composites with lower heat release and smoke production and with enhanced charring ability, it is possible to increase the fire safety in the final appliances.

### 3.4. Mechanical Properties

In the context of the inorganic filler composites, mechanical properties are inevitably concerned besides their flammable characteristics. The high weight fraction of MDH filler additive in polymers can adversely affect some mechanical properties, and often decrease their failure properties [36]. It can be seen from Table 8 that the value of tensile strength decreased from 18.6 to 9.6 MPa when using the MDH to modify the EVA. This reduction in the mechanical properties accounts for the agglomeration of the flame-retardant particles and the interface gaps leading to non-uniform stress distribution in the composites [37]. However, the CeO_2_-containing system shows an improved value of TS (10.1 MPa), and the elongation at break of 202.9%, indicating that the influence of the CeO_2_ structure and particle size can lead to a more uniform dispersion due to an enhanced interfacial interaction (Figure 1d,e) [38]. With the same filler mass ratio, good dispersion will contribute to the significant improvement of Young’s modulus. In this case, Young’s modulus increased significantly from 69 MPa for EM to 76 MPa for EMC3. Its specific tensile curves and the key parameters tendency are all shown in Figure 7. Usually, the tensile strength of the polymeric materials is severely affected when MDH is added as a major flame retardant [39], therefore the ability to reverse this effect by generating an increased tensile strength up to 5% of CeO_2_ represents a very high functionality in this type of material, and this is because the CeO_2_ promotes the mechanical stability of the material by generating better dispersion and combination with the polymer matrix.

### 3.5. UV Aging Performance

#### 3.5.1. Surface Morphology of the Samples after the UV Irradiation

CeO_2_, being one of the most chemically active rare earth oxides, is resistant to ultraviolet light [26]. Along with the incorporation of the rare earth into the formulation of the EVA-MDH composites, we expect to additionally increase the performance of this material as a result of the anti-UV functionalities that the CeO_2_ has inherently. Figure 8 shows the optical microscope images of the pure EVA and EVA/MDH/CeO_2_ composites before and after the UV treatment at different times; for the pure EVA samples, the formation of micro-cracks due to the excessive swelling has led to the displacement of layers in the surface of the polymer after 72 h of UV exposure; the samples containing the pure MDH also showed large cracks (see the red circle area), although the polymer layers were more uniform and the degradation was consistently smaller. In contrast, no significant degradation was observed for the samples containing the CeO_2_ after 72 h of UV irradiation, indicating that the CeO_2_ can effectively delay the diffusion of the UV radiation.

#### 3.5.2. Tensile Test after the UV Exposure

The mechanical test results for the non-exposed and UV-exposed samples are shown in Figure 9. The 72 h of UV exposure had a remarkable deteriorative effect on the mechanical properties of the pure EVA samples, the elongation decreased from 722% until 401% while the tensile strength was also decreased from 18.6 MPa until 8.4 MPa (with a reduction of 54.84%). In the case of the EM samples, a lower decay but significant decrease of the mechanical properties was observed. Namely, the elongation was reduced from 167% until 71.2%, and the tensile strength from 9.6 MPa until 4.9 MPa (with a reduction of 48.96%). As for the EMC3 samples the elongation was reduced from 202% to 94% and the tensile strength slightly decreased from 10.1 MPa to 7.6 MPa (with a reduction of 24.75%). The EMC5 samples showed a more reduced trend, the elongation was reduced slowly from 254% to 192%, and the tensile strength from 9.1 to 7.6 MPa (with a reduction of 16.48%), indicating that the effects of the UV irradiation were significantly decreased with the addition of the CeO_2_. Therefore, it is valuable to note that CeO_2_ could be used effectively as the reinforcement filler to improve the UV-aging resistance of EVA-MDH polymer composites by forming physical barriers that can improve the shielding of the polymer matrix against the UV exposure—the mid-term and long-term degradation of EVA during its effective service usually involves the ultraviolet radiation from sunlight [40], and the systematical enhancement of this functional property is a key solution for improving the EVA polymer composites’ durability.

## 4. Conclusions

EVA composites with MDH and CeO_2_ additives were prepared by hot-melt extrusion and injection molding. The initial flammability tests show the improvement in the LOI values from 18.2% (unmodified EVA) to 40.9% (EM), and after the addition of the rare earth a slight reduction in LOI values was observed—the LOI for EMC3 and EMC5 were 38. 5% and 39.9%, respectively. However, all the modified EVA composites can pass the V-0 rating in the UL-94 vertical burning test. The introduction of the CeO_2_ has created different improvements in the fire properties of the composites measured in the cone calorimetry—a decrease of 59% in the pHRR was obtained for the EM system, while the EMC3 and EMC5 showed a pHRR reduction of 64% and 70%, respectively, and the smoke production was also decreased from 13.9 m^2^ to 7.9 m^2^ (EMC5) and the charring capacity of the samples was enhanced up to 40% (EMC5). The improvements obtained in the flame retardancy of the composites can be related to enhanced activity against the fire in both the condensed and gas phases. The cerium oxide system has presented an interesting advantage in terms of mechanical strength and elongation at break; the tensile strength was improved from 9.6 MPa (EM) to 10.1 MPa (EMC) and the elongation at break was also enhanced from 166% to 254%. The obtained results can reveal that by including non-toxic halogen-free flame retardants in the EVA system, it is possible to improve the functional properties of the EVA composites, and in this case the UV-blocking properties of the CeO_2_ have also led to a higher preservation of the mechanical properties after 72 h of ultraviolet light irradiation. Notably, the multifunctional fillers used in this research work can offer a promising solution in the future developments of high-performance EVA polymer composites. 

## Figures and Tables

**Figure 1 materials-15-05867-f001:**
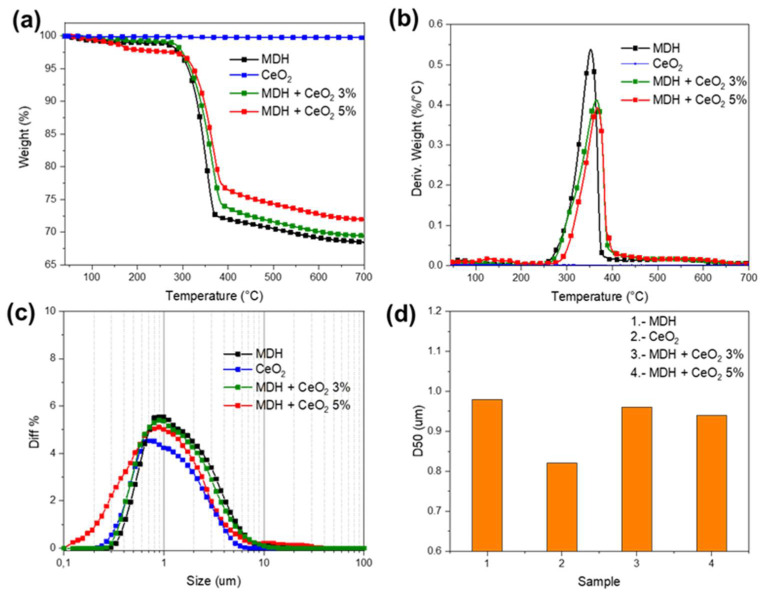
(**a**) TGA (**b**) DTGA (**c**) PSA (**d**) D50 of the flame-retardant fillers.

**Figure 2 materials-15-05867-f002:**
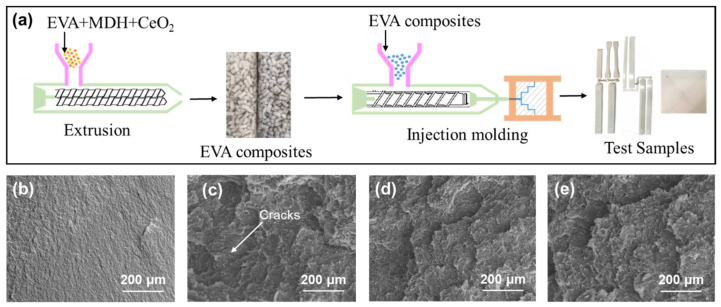
(**a**) Scheme of the samples preparation sequence, and SEM images of the composites cross-section: (**b**) EVA, (**c**) EM, (**d**) EMC3 and (**e**) EMC5.

**Figure 3 materials-15-05867-f003:**
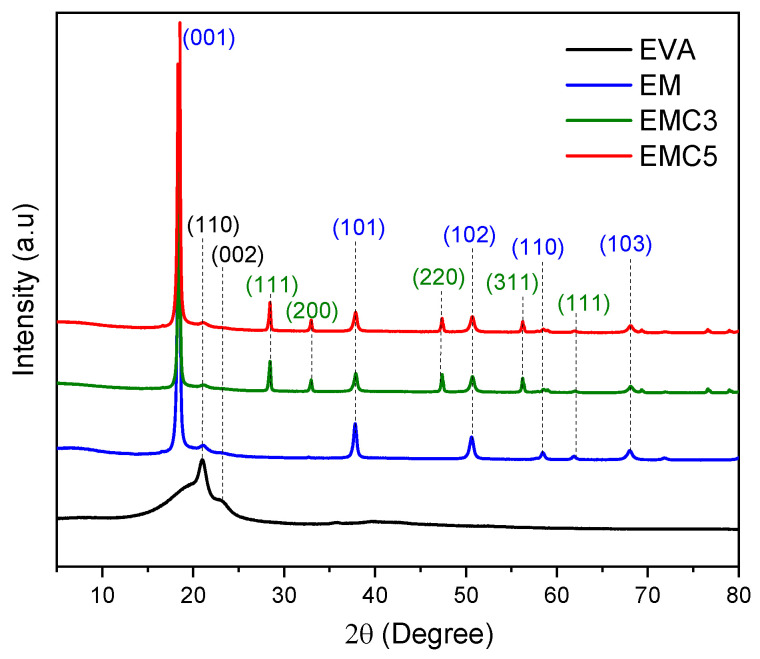
XRD spectra of the EVA-MDH-CeO_2_ composite materials.

**Figure 4 materials-15-05867-f004:**
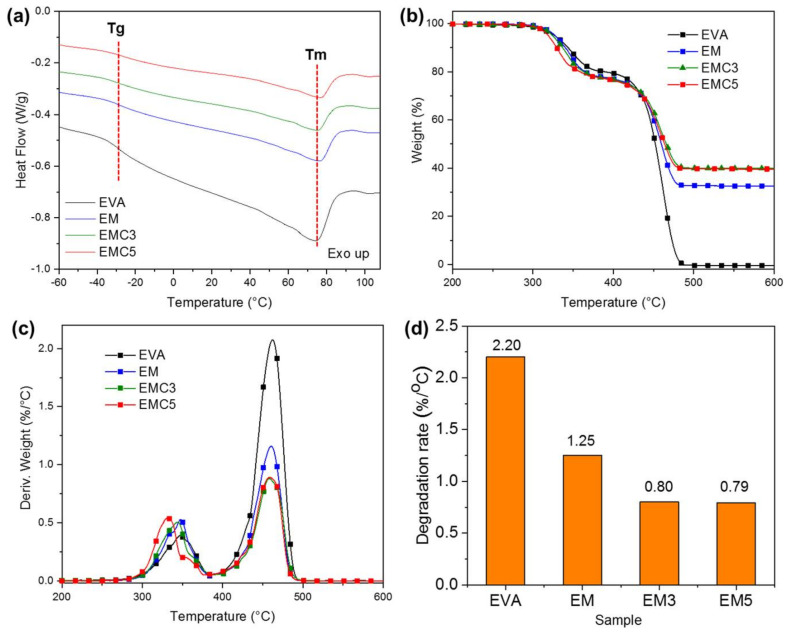
(**a**) DSC curves of the pure EVA, EM, EMC3, and EMC5 composites, (**b**) TGA curves, (**c**) DTGA curves and (**d**) the maximum thermal decomposition rate of the pure EVA, EM, EMC3, and EMC5 composites.

**Figure 5 materials-15-05867-f005:**
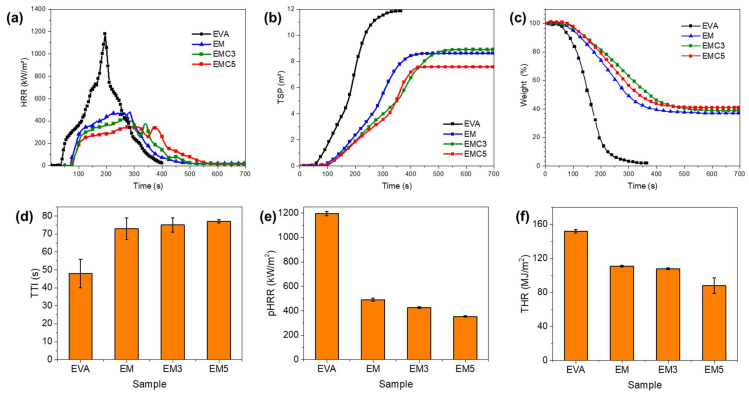
(**a**) HRR curves, (**b**) TSP curves, (**c**) char residue curves, (**d**) TTI, (**e**) pHRR and (**f**) THR of the pure EVA, EM, EMC3, and EMC5 composites.

**Figure 6 materials-15-05867-f006:**
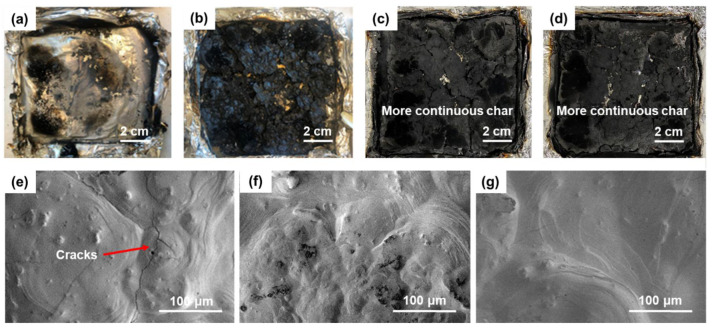
Char residue of (**a**) pure EVA, (**b**) EM, (**c**) EMC3, and (**d**) EMC5 composites after Cone test; SEM images of: (**e**) EM, (**f**) EMC3, and (**g**) EMC5 char residue.

**Figure 7 materials-15-05867-f007:**
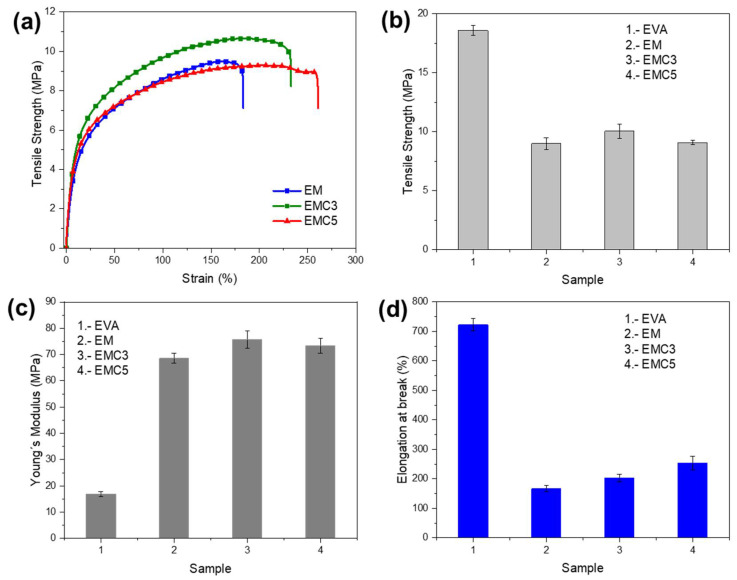
(**a**) Stress-strain curves of the pure EVA, EM, EMC3, and EMC5 composites, (**b**) Tensile strength, (**c**) Young’s Modulus and (**d**) Elongation at break (%) based on the stress strain curves.

**Figure 8 materials-15-05867-f008:**
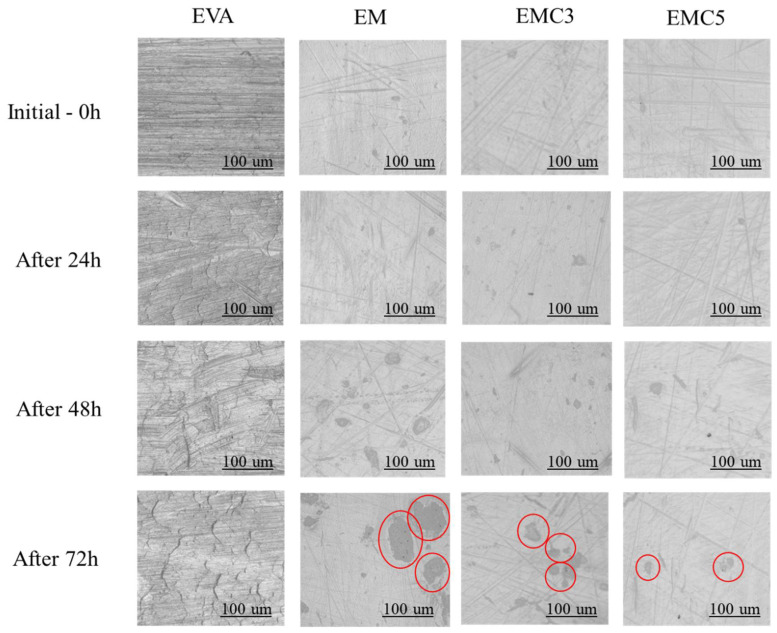
Optical microscope images of the pure EVA and its composites’ surface morphologies before and after exposure to UV, by different times.

**Figure 9 materials-15-05867-f009:**
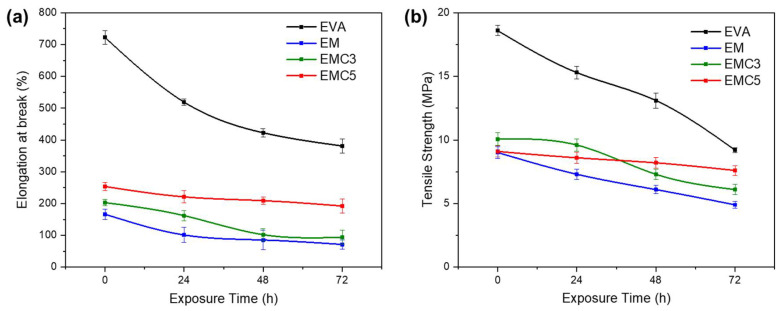
The tensile properties of the samples before and after exposure under UV by different times: (**a**) Elongation at break and (**b**) Tensile strength.

**Table 1 materials-15-05867-t001:** Formulations of EVA Composites (in wt.%).

Sample Name	EVA	MDH	CeO_2_
EVA	100	-	-
EM	45	55	-
EMC3	45	52	3
EMC5	45	50	5

Note: MDH is magnesium hydroxide; CeO_2_ is cerium oxide. EM is magnesium hydroxide-filled EVA; EMC is magnesium hydroxide/cerium oxide-filled EVA.

**Table 2 materials-15-05867-t002:** TGA Results of the Pure and Modified Fillers.

Samples	T-5% (°C)	T-Max (°C)	Der. Weight (%/°C)	Residue @ 700 °C (%)
CeO_2_	-	-	-	-
MDH	312.4	353.1	0.5	68.4
MDH + CeO_2_ 3%	317.5	365.8	0.4	69.5
MDH + CeO_2_ 5%	324.3	367.6	0.4	71.9

**Table 3 materials-15-05867-t003:** PSA Results of the Pure and Modified Fillers.

Sample Formulation	D_50_ (*μ*m)
MDH	0.98
CeO_2_	0.82
MDH + CeO_2_ 3%	0.96
MDH + CeO_2_ 5%	0.94

**Table 4 materials-15-05867-t004:** DSC Results of the composites.

Samples	Tg (°C)	Tm (°C)	Melting Enthalpy (J·g^−1^)
EVA	−28.2	75.1	23.2
EM	−29.5	76.8	13.9
EMC3	−28.9	75.7	11.2
EMC5	−27.5	77.1	11.1

**Table 5 materials-15-05867-t005:** TGA results of the composites.

Samples	T-5% (°C)	T-Max (°C)	Der. Weight (%/°C)	Residue @ 700 °C (in wt.%)
EVA	325.8	460.4	2.1	0.5
EM	327.4	460.5	1.2	32.5
EMC3	324.4	457.8	0.9	39.8
EMC5	319.1	458.4	0.9	39.2

**Table 6 materials-15-05867-t006:** LOI and UL-94 Results of the composites.

Samples	LOI	UL-94			
	(%)	t_1_ (s)	t_2_ (s)	Dripping	Ignition
EVA	18.2	>10	-	Yes	Yes
EM	40.9	1	5	No	No
EMC3	38.5	2	6	No	No
EMC5	39.9	2	4	No	No

**Table 7 materials-15-05867-t007:** Cone calorimetry results of the composites.

Sample Name	TTI (s)	pHRR (kW/m^2^)	pHRR (Red. %)	THR (MJ/m^2^)	AEHC (MJ/kg)	TSP (m^2^)	Char Residue (wt.%)
EVA	48 ± 8	1195 ± 18	-	152 ± 2	35.9 ± 0.1	13.9 ± 3.0	0.6 ± 0.3
EM	73 ± 6	491 ± 13	58.9	111 ± 1	28.6 ± 0.2	8.6 ± 0.6	36.7 ± 0.6
EMC3	75 ± 4	425 ± 7	64.4	108 ± 1	26.3 ± 0.1	8.7 ± 0.5	39.5 ± 0.7
EMC5	77 ± 1	354 ± 6	70.4	88 ± 9	21.8 ± 0.2	7.9 ± 0.4	41.2 ± 0.4

**Table 8 materials-15-05867-t008:** Tensile test results of the composites.

Samples	Elongation at Break (%)	Tensile Strength (MPa)	Young’s Modulus (MPa)
EVA	722 ± 21	18.6 ± 0.4	17 ± 1
EM	167 ± 10	9.6 ± 0.5	69 ± 2
EMC3	203 ± 13	10.1 ± 0.6	76 ± 3
EMC5	254 ± 23	9.1 ± 0.2	73 ± 3

## Data Availability

Not applicable.

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
