# Peer review of "Synergistic Effect of Cerium Oxide for Improving the Fire-Retardant, Mechanical and Ultraviolet-Blocking Properties of EVA/Magnesium Hydroxide Composites"

_materials, 2022, doi:10.3390/ma15175867_

Round 1

Reviewer 1 Report

The manuscript describes the effect of cerium dioxide additive on the EVA/MDH polymer composite properties. The paper is within the scope of the Materials journal, the conclusions are of certain interest for the readership of the journal. The paper is well structured, the material is presented in a consistent way, all conclusions are justified and supported by the results, results are accurately presented, the English is quite appropriate. For the first time, the authors incorporated the CeO2 in EVA/MDH composites and have shown the improvement of the mechanical, fire retardant and ultraviolet blocking properties of the material. The work provides an advance towards the polymer composites and rare earth applications, which makes it publishable in the Materials journal upon some minor corrections.

0. Synergism is mentioned in Title, Abstract and Introduction section. No discussion of the synergistic effect is provided. Why ceria effect is referred to synergistic?

1. Page 2, 69-71 lines. It seems strange to justify the need in the rare earth-related research by the wide availability of the rare earths in China.

2. Page 2, 92 line. Is the diameter or radius of the CeO2 particles specified? What is the shape of the used cerium dioxide particles? Can the morphology of the CeO2 particles influence the properties of the polymer composite?

3. In Figure 2, please mark the XRD reflexes related to each phase.

4. In Figure 3b, the first stage of thermal decomposition is marked as decomposition of MDH, but EVA sample does not contain MDH. TGA data for the pure MDH should be provided or referenced.

5. In Fig. 7, the red circles seem to be drifted from the original positions; some of the circles are outside the area of photograps.

Author Response

The manuscript describes the effect of cerium dioxide additive on the EVA/MDH polymer composite properties. The paper is within the scope of the Materials journal; the conclusions are of certain interest for the readership of the journal. The paper is well structured, the material is presented in a consistent way, all conclusions are justified and supported by the results, results are accurately presented, the English is quite appropriate. For the first time, the authors incorporated the CeO2 in EVA/MDH composites and have shown the improvement of the mechanical, fire retardant and ultraviolet blocking properties of the material. The work provides an advance towards the polymer composites and rare earth applications, which makes it publishable in the Materials journal upon some minor corrections:

Comment 1-0. Synergism is mentioned in Title, Abstract and Introduction section. No discussion of the synergistic effect is provided. Why ceria effect is referred to synergistic?

Response: The combination of ceria with magnesium hydroxide enables the formation of a material with properties that the single hydroxide does not possess. The following aspects reveal a more insightful approach of the enhancements in the presented polymer composites:

  • The TGA test (now included in section 3.1.1) was performed for the pure MDH, pure CeO2 and for the modified MDH fillers with 3% and 5% of CeO2, the obtained results indicate that after the addition of the rare earth oxide, the initial decomposition temperature of the flame retardant system was increased and the charring capacity was also improved by promoting the formation of a more continuous and more compact barrier layer, therefore a suitable amount of CeO2 can play a key role in the thermal stability and char-forming behavior of the EVA/MDH composites. This decomposition behavior is in agreement with the TGA results discussed elsewhere [8].

Table 2. TGA Results of the pure and modified fillers.

Samples

T-5% (°C)

T-Max(°C)

Der. Weight  (%/°C)

Residue @ 700 °C (%)

CeO2

-

-

-

-

MDH

312.4

353.1

0.5

68.4

MDH + CeO2 3%

317.5

365.8

0.4

69.5

MDH + CeO2 5%

324.3

367.6

0.4

71.9

Figure 1. (a) TGA (b) DTGA (c) PSA (d) D50 of the flame retardant fillers.

  • The additive dispersion through the EVA polymer matrix is a fundamental property for this research work, due to the existence of a high proportion of inorganic particles. The addition of the CeO2 (with particle size: 0.82 m) into the MDH represents an improvement in the dispersion and interfacial interaction levels of the composites. The fracture roughness of the polymer composites reflects the dispersion level and interfacial interaction to some degree [31]. The morphology and structural analysis of the EVA/MDH composites revealed in section 3.1.2, indicates that after the addition of the rare earth a smoother surface with smaller gaps between the filler and polymer matrix was obtained, which can help to explain the improvements obtained in mechanical properties.
  • The addition of ceria into the EVA/MDH has led to the improvements of the fire resistance and mechanical properties, in addition, it was also possible to improve the surface and mechanical properties preservation after UV-light exposure, due to the ultraviolet blocking capacity of the rare earth. Although the MDH has the mayor amount participation in this flame retardant system, a considerable low amount of the rare earth can remarkably improve the functional properties performance of the EVA composites.

Comment 1-1. Page 2, 69-71 lines. It seems strange to justify the need in the rare earth-related research by the wide availability of the rare earths in China.

Response: By following the reviewer comment, we have replaced some sentences and now it reads: The growth of rare earth elements into technology advancement, the ecology, and economic domains has resulted in a major increase in global demands. Over the next decade, global demand for cars, electronic goods, energy-efficient lighting, and catalysts is predicted to surge. In order to encourage future advances in this field it is necessary to perform further studies for its applications, and flame retardant synergistic effect is one of them. [33]

Comment 1-2. Page 2, 92 line. Is the diameter or radius of the CeO2 particles specified? What is the shape of the used cerium dioxide particles? Can the morphology of the CeO2 particles influence the properties of the polymer composite?

Response: The diffraction peaks obtained in the XRD test suggested that these particles have the cubic fluorite-structured cerium oxide with diffraction rings of (111), (200), (220), and (311). The particle size of the fillers was tested in our lab, the PSA results are revealed in Table 3 (now included in 3.1.1 section), the results indicate that the pure MDH has a standard size of 0.98 m, while the CeO2 was sized at 0.82 m, the influence of the rare earth on the MDH flame retardant indicates that for the fillers containing 3 wt. % of the CeO2 the particle size obtained was of 0.96 m, and for the fillers containing 5 wt. % of CeO2 there was a 0.94 m particle size obtained.

Table 3. PSA Results of the pure and modified fillers.

 Sample formulation

D50 (m)

MDH

0.98

CeO2

0.82

MDH + CeO2 3%

0.96

MDH + CeO2 5%

0.94

Due to the high loading of inorganic particles (most of them are MDH), in this case, we think that the smaller size of the CeO2 particles is more suitable to improve the interfacial performance of the composites, as an alternative of the particles morphology which can be similar for all the formulations of this research work.

Comment 1-3. In Figure 2, please mark the XRD reflexes related to each phase.

Response: By following the reviewer comment, the diffraction peaks were indexed and included in Figure 3 (section 3.1.3)

Figure 3. XRD spectra of the EVA-MDH-CeO2 composite materials.

Comment 1-4. In Figure 3b, the first stage of thermal decomposition is marked as decomposition of MDH, but EVA sample does not contain MDH. TGA data for the pure MDH should be provided or referenced.

Response: We have removed some indication words from Figure 4. The TGA of pure MDH was included in Figure 1. (section 3.1.1)

Comment 1-5. In Fig. 7, the red circles seem to be drifted from the original positions; some of the circles are outside the area of photographs.

Response: After revision, the red circles were moved to the original position in Figure 8.

Reviewer 2 Report

The work presented in this manuscript is interesting and discsuss important topic and worthy to be publish. However, there are few points need to be addressed.

1. The abstract need to enriched with novelty.

2. Introduction section need to be enriched with some reported metal oxide- and other organic nanoparticles  as flame retardant fillers and I recommend citation of the few reported reports 

Journal of Thermal Analysis and Calorimetry 127 (2017) 2273-2282

Progress in Organic Coatings 110 (2017)204-209

3. Authors have to explain  what is exactly the role of CeO2 in flame retardancy action.

4. Authors have to insert effective heat of combustion data from cone data and explain

Author Response

The work presented in this manuscript is interesting and discuss important topic and

worthy to be publish. However, there are few points need to be addressed.

Comment 2-1. The abstract need to enriched with novelty.

Response: Following the reviewer comment, some sentences were replaced in the abstract, and now it reads: “Rare earth oxide particles have received important attention in the last years, due to the wide diversity of promising applications, the need for this kind of materials is predicted to expand as the requirements to use the current resources becomes more demanding. In this work, cerium oxide (CeO2) was introduced in the ethylene-vinyl acetate (EVA)/magnesium hydroxide (MDH) composites for enhancing the flame retardancy… … Notably, the combination of CeO2 with MDH is a novel and simple method to improve the filler polymer interaction and dispersion which resulted in the improvement of the mechanical properties, flame retardancy and the anti-ultraviolet aging performance of the composites”.

Comment 2-2. Introduction section need to be enriched with some reported metal oxide- and other organic nanoparticles as flame retardant fillers and I recommend citation of the few reported reports

Journal of Thermal Analysis and Calorimetry 127 (2017) 2273-2282

Progress in Organic Coatings 110 (2017)204-209

Response: By following the reviewer comment, the introduction section was modified and now it reads: “The compatibility effects of flame retardant fillers for producing uniform dispersed polymer composites has been previously studied, by using a phosphorous-based polypyrrole nanoparticles it was possible to obtain significant improvements in the thermal stability and flame resistance of acrylonitrile-butadiene-styrene composites…” And the recommended articles are also properly cited in the revised manuscript.

Comment 2-3. Authors have to explain what is exactly the role of CeO2 in flame retardancy action.

Response: Following the reviewer comment, some sentences were replaced in section 3.3 and “The effect of the CeO2 in the flame retardancy of the composites is included as section 3.1.1”. Now it reads: “The fundamental role of CeO2 in the flame retardancy enhancements is assigned to the yielding of a larger amount of solid inorganic residues that serve as a barrier to heat and mass transfer between the pyrolysis zone and the underlying polymer…”

Comment 2-3. Authors have to insert effective heat of combustion data from cone data and explain.

Response: By following the reviewer comment, AEHC was included in section 3.1.1, now it reads: “…The effective heat of combustion (EHC), defined as the ratio of heat release to mass loss at a certain time during combustion, can exactly reveal the burning degree of volatiles in gas phase [36-39]. As shown in Table 6, average EHC (AEHC) was decreased from 35.9 MJ/kg for EVA to 28.6 MJ/kg for EM, and further decreased to 26.3 and 21.8 MJ/kg with the addition of3 and 5 wt. % of cerium oxide respectively; the enhanced EHC performance was attributed to the increase of concentration of noncombustible compounds in the gas phase and the higher char residue formed by the inorganic fillers action…”

Table 6. Cone calorimetry results of the composites

Sample Name

TTI (s)

pHRR (kW/m2)

pHRR (Red. %)

THR (MJ/m2)

AEHC (MJ/kg)

TSP

(m2)

Char residue (wt.%)

EVA

48 ± 8

1195 ± 18

-

152 ± 2

35.9 ± 0.1

13.9 ± 3.0

  0.6 ± 0.3

EM

73 ± 6

  491 ± 13

58.9

111 ± 1

28.6 ± 0.2

  8.6 ± 0.6

36.7 ± 0.6

EMC3

75 ± 4

425 ± 7

64.4

108 ± 1

26.3 ± 0.1

  8.7 ± 0.5

39.5 ± 0.7

EMC5

77 ± 1

354 ± 6

70.4

  88 ± 9

21.8 ± 0.2

  7.9 ± 0.4

41.2 ± 0.4
